# Social Collective Attack Model and Procedures for Large-Scale Cyber-Physical Systems

**DOI:** 10.3390/s21030991

**Published:** 2021-02-02

**Authors:** Peidong Zhu, Peng Xun, Yifan Hu, Yinqiao Xiong

**Affiliations:** 1Department of Electronic Information and Electrical Engineering, Changsha University, Changsha 410022, China; zpd@ccsu.edu.cn (P.Z.); yq.xiong@ccsu.edu.cn (Y.X.); 2College of Computer, National University of Defense Technology, Changsha 410073, China; huyifan17@nudt.edu.cn

**Keywords:** cyber-physical system, security, collective behavior, social users, threat model, disinformation

## Abstract

A large-scale Cyber-Physical System (CPS) such as a smart grid usually provides service to a vast number of users as a public utility. Security is one of the most vital aspects in such critical infrastructures. The existing CPS security usually considers the attack from the information domain to the physical domain, such as injecting false data to damage sensing. Social Collective Attack on CPS (SCAC) is proposed as a new kind of attack that intrudes into the social domain and manipulates the collective behavior of social users to disrupt the physical subsystem. To provide a systematic description framework for such threats, we extend MITRE ATT&CK, the most used cyber adversary behavior modeling framework, to cover social, cyber, and physical domains. We discuss how the disinformation may be constructed and eventually leads to physical system malfunction through the social-cyber-physical interfaces, and we analyze how the adversaries launch disinformation attacks to better manipulate collective behavior. Finally, simulation analysis of SCAC in a smart grid is provided to demonstrate the possibility of such an attack.

## 1. Introduction

CPSs (Cyber-Physical Systems) are systems that connect the physical world with the digital. The typical CPSs are smart grids, intelligent transportation systems, and other modern engineered systems. Many of them work as critical infrastructures and provide services to millions of users. The security of such large-scale CPSs is becoming increasingly important [1,2]. Their complex structure and numerous nodes are likely to lead to cyber vulnerabilities, which may seriously affect the quality of life of a large number of users.

In existing CPSs, the security basically considers the attack from the information domain to the physical domain [3,4], such as injecting false data to damage sensing [5] or illegally manipulating the command control [6]. On the other hand, with the increase of operations in social space for large-scale CPSs, social users can also greatly influence the CPS, and the social attribute of the CPS is increasingly obvious, thus forming a CPSS (Cyber-Physical-Social System) [7]. Currently, the social character of users can be utilized to improve the service of large-scale CPSs. For example, Reference [8] used the social relationship of users to improve energy harvesting and communication between devices in the physical world. Reference [9] proposed a novel method based on the information collected from social users to detect and locate power outages. Besides that, to better provide services and implement intelligent systems, some mechanisms like regulation based on sensory data from users [10] and demand-response services [11] that provide better services based on user demands are widely used. However, the influence of users is like a double-edged sword. The human factor is also often regarded as the weakest link in cybersecurity [12] because of the uncertainty of human behavior [13]. Manipulating a large number of users may be an attack entrance to degrade performance or even destroy large-scale CPSs. Currently, such attacks caused by manipulating social users for large-scale CPSs is a research area that has seen limited work.

The demand-response services in a power grid [11] can be maliciously exploited to disrupt the system stability. In [14], false price modification was injected into smart meters, and the power users changed their load demand to respond to the changing prices. The abnormal load variation led to system instability. In [15], the IoT devices of power appliances were manipulated, and the large number of electrical synchronous switches affected the stability of the whole power grid. Although [14,15] are good examples that cause the power grid instability through the intrusion into the end systems of power users, the attacks were initiated from the cyber domain.

In order to achieve the desired attack effect in [14,15], the attacker needs to manipulate a large number of cyber elements as entrance points, thus increasing the difficulty and cost of intrusion and leaving traces that can be easily detected. Therefore, attackers try to explore new attack methods, such as whether they can take advantage of the large-scale online social network and impact electrical usage by influencing a large number of social network users who are also power users. Therefore, a new attack mode with the attack entry in the social domain is created and proposed in this paper.

In this paper, we focus on the new kind of attack for large-scale CPSs called Social Collective Attack on CPS (SCAC), which maliciously manipulates the collective behavior of a large number of users by directly intruding into the social domain to disrupt the CPS. In the subsequent sections, we will introduce our framework for SCAC, which is launched by disinformation diffusion on social media to disrupt the physical functioning of the whole system, and we discuss various aspects of its implementation. Disinformation is false information that is given deliberately to mislead the public. Our main contributions in this paper can be summarized as follows:We introduce a model of social collective attack on physical systems, which makes full use of the characteristics of cyber-physical-social interactions and the integration of infrastructures such as smart grids.

The existing CPS security basically considers the attack from the information domain to the physical domain. The security of social elements in large cyber-physical systems is left unexplored, including the impact of the social domain. Generally, people think it is impossible to launch attacks on the physical domain from the social domain, as the two domains are usually separated by the cyber domain. What is more, for example, as for the security of the smart grid, it is easy to understand that the power plant operators of the smart grid may directly endanger the normal operation of the power grid, while the end users can hardly directly affect the stable operation of the power system. However, the SCAC model implements the latter.

We extend MITRE ATT&CK [16], the most used cyber adversary behavior modeling framework to cover cyber, physical, and social domains. In other words, we provide a systematic description framework for security threats that are launched from the social domain, penetrate through the cyber domain, and target physical domains.

MITRE ATT&CK is used worldwide across multiple disciplines including intrusion detection, threat hunting, security engineering, threat intelligence, red teaming, and risk management. MITRE ATT&CK is the well-known framework for cyber adversary behavior, reflecting the various phases of an adversary’s attack lifecycle and the platforms they are known to target. It was created in 2013 and used initially only in the cyber domain.

There is one significant extension of the ATT&CK framework to the physical domain, i.e., ICS ATT&CK [17], the extension for Industrial Control Systems’ attack, which characterizes and describes the actions of an adversary who seeks to cause a negative effect on the physical system. It is extended with additional tactics and techniques to address adversarial behavior in OT (Operational Technology) networks that can disrupt or destruct physical subsystems. The described kill chain starts from the cyber domain and targets the physical domain.

ATT&CK has no consideration of social domain behaviors, while AMITT (Adversarial Misinformation and Influence Tactics and Techniques) [18] enables analysts to quickly describe adversarial misinformation in the social domain. AMITT is a work of the MisinfoSec Working Group to adapt information security practices to help track and counter misinformation. The described kill chain starts from the social domain, and the generated misinformation spreads on the basis of the information system (cyber domain). AMITT does not consider including the physical domain. however, our SCAC is launched from the social domain and penetrates through the cyber domain to the physical domain, with the objective of affecting the physical system’s functioning. The new kill chain combines AMITT and ICS ATT&CK and extends ICS ATT&CK to include attacks initiated from the social domain.

We give an extensive analysis of the implementation of SCAC in a smart grid called Drastic Demand Change (DDC) attack, which manipulates a large number of users by disinformation to modify their electricity consumption behavior, and the sudden change of power load demand leads to the instability of the power grid. The simulation and experimental results show that the DDC attack can cause the duration of a frequency deviation to exceed the threshold and lead to the disconnection of power generators from the grid. As for the two methods to realize DDC attack, the reverse demands attack can achieve a better impact than the fast attack.

The rest of the paper is organized as follows. In Section 2, we introduce the related work. We introduce and analyze the social collective attack model (SCAC) and procedures in Section 3. In Section 4, we describe the disinformation attack and its evaluation in relatively formal language. We present our case study about such an attack in the context of a smart grid in Section 5. We conclude the paper in Section 6.

## 2. Related Work

SCAC exploits the interactions of the cyber, physical, and social domains. In this section, we first summarize the CPS security models that mainly consider the threats from the cyber domain to the physical domain. Then, we describe the works related to attacks initiated from social domains especially through disinformation diffusion. One important implementation of SCAC is in the power grid, which abuses the demand response mechanism, so we provide some background knowledge for readers to understand the possibility of SCAC and compare SCAC with other demand-side manipulation attacks.

### 2.1. Traditional Security Model of CPS

Many previous works discussed security issues in CPSs. The attacks are usually launched from the cyber domain to disrupt the physical domain [3]. Reference [3] introduced the possible attack behavior and the vulnerabilities of CPS combining the cyber domain with the physical system. Reference [4] reviewed the threats for Supervisory Control and Data Acquisition (SCADA) systems. In [5], a false data injection attack was described, which first compromises the sensors of the physical system and injects some carefully constructed bad data into the sensors, thus leading to the controllers falsely evaluating the physical state of the system and making incorrect decisions. In [6], the false sequential logic attack was introduced, where an attacker directly compromises the SCADA controller and constructs false logic control commands that would lead to direct damage to the physical system.

On the other side, physical system sensing can also be used to prepare for cyberattack. In [19], the authors collected acoustical signals and used data mining to restore information in the cyber domain. However, the impact on the social domain was not explored and is largely left unexplored.

### 2.2. Attacks Initiated from Social Networks

Misinformation is false or inaccurate information. Disinformation is deliberately deceptive information. Misinformation or disinformation can cause cybersecurity events. Within two hours, the spread of the event of WeChat Open Class PRO link stealing caused the panic of millions of users to unbind WeChat bundled bank cards or withdraw cash, which led almost to the paralysis of the Tencent server [20]. In this case, the behavior of the social domain leads to the security problem of the cyber domain.

Some recent works have shed some light on CPS security issues incorporating the social domain. For instance, Reference [21] presented a general framework for analyzing the security of critical infrastructure in terms of three different interaction layers: physical layer, cyber layer, and social layer, where the social layer refers to the actions of the operators. In the real world, attack examples do exist involving or through operators, e.g., the Stuxnet attack [22]. In [23], the reliability aspects were considered when bad sensory data are fed into the system from a large number of users in a crowd sensing scenario. With the increasing popularity of social networks, the role of social users in CPSs is becoming more prevalent. Reference [24] analyzed this new problem by modeling a social network-coupled smart grid and investigating its vulnerability to false pricing attacks in the social network. In this case, the intrusion of the social domain led to the security problem of the physical domain.

### 2.3. Power System Attacks by Demand-Side Manipulation

A smart grid is a cyber-physical system. The underlying information system facilitates interactions between customers and the power system, and the physical system is integrated with sensing, communication, computing, and control modules. Demand response is an enabling technology to achieve energy efficiency in a smart grid. A simple demand response model is depicted in Figure 1.

Demand response enables dynamic modification of electricity demand into the operations of power grids. There are two basic modes of power load manipulation at the end user side, i.e., direct load control and indirect load control.

(1)Direct load control and related attack:

Some appliances can be remotely shut down by the company with the permission of the consumers, or customers voluntarily shut down appliances after receiving a notice from the utilities. They can receive a reward on their electricity bills for this participation [25]. The direct load control capability may be abused by attackers. For example, malwares can be injected into the central controller or smart home applications, and the consumer appliances are maliciously manipulated [19]. The botnet of compromised power-consuming IoT devices could be commanded to switch on or off at the same time, abruptly increasing or decreasing power demands and creating an imbalance between power supply and demand with dramatic effects [15].

Figure 2 depicts the model of the above attack, initiated from the cyber domain (and then probably spread in the cyber domain), causing abnormal synchronism in the load and then cascading failures in the physical system. The information system is the first step attack target, and traces of the attack are left in the CPS, which could be detected easily. Attackers invade IoT devices in the cyber domain and directly control the electrical appliances in the physical domain, and the process can be characterized as “cyber→physical”.

(2)Indirect load control and related attack:

Indirect load control in demand response is implemented usually by pricing-based approaches. By injecting false price signals, an attacker can plan to simultaneously change the energy consumption of a large number of appliances and cause improper changes in the load profile, which eventually leads to transmission line damage in large areas due to overload [14]. In [26], if the total demand decreases from the utility company, a part of the available power will be wasted; on the other hand, if the demand increases more than the available power, overload, and even black-out, may occur.

Figure 3 depicts the attack model triggered by false price data injection in the cyber domain. The power users may read the price information from smart meters and reschedule their consumption to maximize the anticipated benefit, and the process can be characterized as “cyber→social→physical”.

(3)Social collective attack on the CPS:

SCAC, proposed in this paper, can be implemented in the power grid. Figure 4 depicts the simplified model of an attack launched from the social domain by faking messages related to electricity price or grid running status, which are disseminated in the social space and induce some users to operate the electrical appliances abnormally. When a large number of users are tricked and take actions that have a negative impact on the physical system, a social collective attack is implemented.

Since the power users can be tricked by different kinds of messages including the faked commands from the utility company and electricity prices, SCAC can comprehensively use direct and indirect load control modes. Compared to the attacks described in [14,15], SCAC is characterized as “social→physical”.

## 3. Attack Model and Procedures

In this section, we first introduce the system model of the CPS combined with the social domain. Then, we describe the SCAC model and major procedures, which implement the attacks on the physical system initiated from disinformation propagation on social media.

### 3.1. System Model of CPS Combined with the Social Domain

CPS is a term coined by the NSF(National Science Foundation) in the United States in 2006, used to describe systems that connected the physical world with the digital one. For example, a power grid is initially a physical system, but with the integration of an information system, which helps to monitor and control the power elements, the smart grid comes into being as a cyber-physical system. The CPS has been an active research topic for more than a decade [1,2].

With the development of CPS applications, especially Industry 4.0 or the Industrial Internet, human factors have been taken into consideration gradually. The interaction between the humans and the machines and the physical environments is getting more attention. Thus, the CPSS [7] emerges as the integration of the computing, physical, and human resources and with the coordination among the cyber, physical, and social worlds. For example, in the case of a smart grid, power users and human operators in the power plant are elements of the social domain.

We are among the first researchers that have recognized the importance of social elements in CPSs and have extended the basic CPS model to the CPSnet (Cyber-Physical-Social network) model. The work in this manuscript really started in 2017; at that time, the idea of launching an attack from social users to disrupt the physical power grid was quite new, and most people thought it impossible. Now, our research work focuses on the new CPSnet model or CPSS model, which regards humans as a part of the system and puts humans in the loop.

The smart grid extends the traditional electric system from a producer-controlled physical system to a cyber-physical system that supports consumers’ active engagement in energy management. Human factors in the smart grid also include power consumers, as well as power operators, and both of them usually interact with the physical system through the information system. Power users can communicate with each other on social media to share information and receive notifications from the power company. A typical model for a large-scale CPS combined with the social domain is shown in Figure 5, which forms a cyber-physical-social system.

Next, we describe the functions of the three domains and the interactions among them.

Physical domain: The physical domain operates in the physical world and provides services to the end users. Elements in the physical domain include devices in the engineering domain, which interact with the cyber domain through sensors, actuators, and controllers. Sensors act as detectors to capture the physical data and transmit data to the information system via communication channels. Actuators execute commands from controllers and operate directly in the physical world. Controllers receive commands from the information system and convert semantic commands into signals that can be understood by the actuators.Cyber domain: The cyber domain is comprised of the information system that transmits the state of the physical system and sends control commands to the engineering equipment, and social media is also based on the information system that carries the communications among end users. For electrical appliances, the smart meters and smart home apps work mainly in the cyber domain and interact with the physical systems.Social domain: There mainly exist two kinds of social roles: operators and CPS users. In this paper, we mainly pay attention to CPS users. The users can get service from the CPS and provide feedback to the system. Power users’ behaviors have an influence on the operation of the physical system, and their thoughts can be influenced by the communication among people on social media.

The three domains are integrated as a CPSS. The cyber domain supports the communication and interconnection among physical components and among social users; as the interface between the physical domain and the social domain, smart meters and smart home applications detect and display the physical system status and show it to the users, or the users control the physical system through them.

### 3.2. Model and Steps of Social Collective Attack on CPS

Social media manipulation and misinformation campaigns have increased globally in recent years [27]. The structure and propagation patterns of misinformation incidents have many similarities to those seen in information security. AMITT [18] is a work b they MisinfoSec Working Group to adapt information security practices to help track and counter misinformation, which enables analysts to quickly describe adversarial misinformation. AMITT is to MisinfoSec as MITRE ATT&CK [16] is to InfoSec. The ATT&CK matrix includes phases, tasks, and techniques. The operational phases of misinformation incidents include: Initial access, create artifacts, insert theme, amplify message, command and control [18]. SCAC will extend the above phases with the ultimate attack objective of an impact on the physical system.

ICS ATT&CK [17] is a knowledge base for industrial control systems’ attack, which characterizes and describes the actions of an adversary who seeks to cause a negative effect on the physical system. SCAC will run through and connect the corresponding components of AMITT and ICS ATT&CK, as well as use the TTP (Tactics, Techniques, and Procedures) initiated from the social domain, through the cyber domain, finally affecting the physical domain. In some sense, SCAC can be called AMITT with a specific physical impact on the CPS as the final objective, or an ICS ATT&CK that is launched from social media.

In order to generate the desired attack result on the physical system, one step before initial access to social media is required, which is an investigation into the targeted physical system; disinformation is forged related to, e.g., electricity price or a power facility failure event, which is injected, propagated, and amplified on social media; some users will believe in the fake message and operate their electrical appliances as the attacker expects; the abnormal operation of a large number of loads will cause power system variation and even failure. We design the following SCAC steps as depicted in Figure 6: (1) reconnaissance and planning; (2) disinformation fabrication; (3) disinformation delivery (propagation and amplification on social media); (4) disinformation exploitation (command and control, impact on the physical system); (5) evaluation and calibration (feedback and further action). These steps are not a simple combination of AMITT and ICS ATT&CK, but regard the target system as a complete CPSS at each phase.

#### 3.2.1. Reconnaissance and Planning

The modes of CPS attack can be divided into two categories: mass destruction and precision strike. The attack is initially launched from social media; the generation of disinformation messages needs to be based on the characteristics of the physical system and the social network. Therefore, reconnaissance is necessary to detect, collect, and analyze the structure and characteristics of the physical system and social media. In terms of smart grid disruption, knowledge acquiring may be related to the social network, power system, coupling between the social and power domain, and the network structure model, the propagation or cascading model.

Sometimes, attackers can directly sense the state of the CPS by connecting with the physical system and do not need to intrude into the CPS. For example, when attackers hope to sense the frequency of power grid and then increase or decrease the load to disrupt generators, a machine can be used directly to connect with terminal electrical appliances to sense the frequency of the power grid system [28]. In most of the cases, however, the state of the physical system cannot be directly sensed. Attackers need to collect information by intruding into the information system to evaluate the state of the physical system, which is possible, as described in [3]. At the same time, because attackers only hope to get information about the CPS and do not execute any operation, sensing of the CPS state is difficult to detect.

Goal planning includes the formulation of specific action routes. AMITT’s target is people’s minds, while SCAC takes the physical system as the final target. The strategy defines the desired end state, which is the set of required conditions needed to achieve all the objectives. The objectives should be clearly defined, measurable, and achievable. To achieve the objectives, tasks involving message fabrication and dissemination should be well designed. How the messages affect the targeted users and whether the physical system can be disrupted may be rehearsed in advance and evaluated quantitatively using the knowledge about the social network, the power grid, and the coupling between users and electrical appliances.

#### 3.2.2. Disinformation Fabrication

Disinformation fabrication is the weaponization of social messages. The goal of these messages is to affect the physical system accurately, but its first step is implemented by manipulating the power grid users effectively, i.e., they can be effectively spread on social media and recognized by the corresponding users. Generally, the techniques of content development include distorting facts, conspiracy narratives, leaking altered documents, and others [18]. In the case of the smart grid, fake messages may be related to price information or system notification, which will induce the power users on social media to change the electric loads as the adversaries expect.

To induce changes in the load profiles of individual users and eventually cause major alterations in the load profile of the entire network, the messages should be meticulously designed, and perhaps, a series of messages is needed to induce the users to approach and realize the attacker’s objective gradually. Another requirement is the credibility of the messages, which ensures a certain number of users or specific users are influenced and operate their appliances as expected.

Strategies: mass destruction is for greedy attack, and microtargeting is for precision targeted attack.

Tactics: forgery of price information, forgery of line/system fault information and outage notice, false maintenance notice, and so on.

Techniques: price-based approach, incentive-based approach, loss-avoidance approach, and environment-aware approach. These techniques make full use of the psychology and value orientation of power users. Section 3.3 provides the procedures to implement the “Drastic Demand Change” (DDC) attack by using these techniques.

(1)Price-based approach:

For example [29], if a message is fabricated about the lowering of the electricity price for some time during the day, this would lead to users increasing their load during these intervals, and sudden spikes in the demand may occur, which will probably lead to system instability.

(2)Incentive-based approach:

There is a variety of forms of incentive-based approaches [30] that can be used to create the messages: (i) direct load controls, where customers are informed about receiving some payments for allowing the utilities to control the operation of some equipment during specific hours of the day and season; (ii) emergency demand response, where customers can receive payments for voluntarily reducing their loads during the claimed emergency periods in the name of increasing the grid reliability; (iii) interruptible/curtailable programs, where customers are told they can receive discounts or credits for allowing the utilities to reduce their loads when requested.

(3)Loss-avoidance approach:

The loss-avoidance approach can be considered as another category of incentive-based approach, which notifies the users that if they do not change their load demand, some loss will be caused. For example, penalties will be enforced if the customers cannot perform the desired curtailments; faked system maintenance notices and failure notifications also belong to this category.

(4)Environment-aware approach:

The attackers can abuse the users’ environmental awareness. For instance, the Earth hour [31] is an activity that requests people to switch off the lights at homes and businesses for an hour at 8:30 p.m. usually on the last Saturday of March. Attackers could imitate a similar campaign at an appropriate time or induce people to change their original electricity use behavior in the name of environmental protection and green development.

It should be noted that SCAC manipulates electrical appliances through users in the social domain. Different from false price injection in the information domain, especially Real-Time Pricing (RTP) attack, SCAC cannot automatically control appliances in a short time scale [14] because of the lag and uncertainty of human response, while RTP may update energy prices typically every 5 or 15 min.

#### 3.2.3. Disinformation Propagation and Amplification on Social Media

After creating the disinformation, the next step is the delivery through social media. The goal is to disseminate the disinformation to those users who will probably accept it, and their following operation will lead to the expected physical effect. The preparation work includes: (1) select infrastructure, develop people, and develop the network; (2) publish the content, inject content into the media, and obscure the origination; (3) amplify the propagation and mislead the public.

To implement effective and efficient dissemination, reconnaissance and investigation of the social network are necessary, including the distribution of social users, their tendency to believe, and coupling with the physical system. It is an optimization problem to select the set of social users whose manipulation could lead to maximum disruption of the smart grid. As [29] defined, the social network is $G_{S}=\left(V_{S},E_{S},p\right)$, the power network is $G_{P}=\left(V_{P},E_{P}\right)$, and $E_{PS}$ is the edge set connecting the above two graphs; the problem can be defined as identifying *k* nodes in $G_{S}$ whose activation would lead to the maximum number of failed/disconnected nodes in $G_{P}$ based on the disinformation attack.

If there is not enough knowledge about the social media platform, the greedy social attack strategy [29] may be applied, whose effectiveness is not assured. Reference [18] listed the tasks and corresponding techniques that can also be used in SCAC, for example repeat messaging with bots or by friends to quickly develop the network and expand the diffusion capability, impersonate or hijack the account number of the power company, or malicious use of some VIPs to increase the credibility of messages.

The user behavior model is very important in disinformation delivery [32]. Figure 7 depicts the user behaviors in SCAC after receiving the message. The user will judge the message’s authenticity and credibility. If he/she believes in the content, the user may forward the message to other power users who may or may not be members of the social media platform; some of the believers may take action. In the chain of propagation and attack, there are many uncertainties in each link, so a model of the stochastic process or probability is necessary to describe the user behaviors.

Many previous works have demonstrated that disinformation can be widely propagated on social media [33] and greatly influence the collective behavior of users [34]. With the change of time, disinformation can be propagated quickly with the interactions among users. If a user chooses to believe in the disinformation, his/her electricity demands may change according to the content. Once enough people believe in the disinformation, demands can be changed drastically by attackers.

#### 3.2.4. Disinformation Exploitation

Disinformation in different fields has different uses [32]. Governments can use disinformation to exercise control over groups of people. Businesses can use disinformation to maintain or repair their own reputation or to damage the reputation of a competitor. In SCAC, as Figure 4 describes, the vulnerability of social media and the smart grid can be exploited, and when the disinformation is diffused as expected, the command and control are about to start. The attack will penetrate from the social domain, through the cyber domain, and reach the physical domain to change the electricity load. When the difference between the manipulated demand and the actual demand is large, the state of the physical system may have an exceptional transition.

#### 3.2.5. Evaluation and Calibration

The final step is to evaluate the attack’s effects on the physical system. If the whole objective is achieved, this attack is successful; if a phase’s objective in a series of attacks is achieved, a new phase of attack can be launched; if the attack’s effect deviates from the expectation, adjustment and calibration are necessary, i.e., the SACAis a feedback-based iterative attack. The adjustment needs to take place in the steps that may lead to the deviation.

From the description of the attack model, we can find that the attack process has less possibility to leave traces in the physical system and cyber domain, which means that this kind of attack is difficult to detect by traditional methods based on data from the information system and the physical system.

### 3.3. Disinformation Fabrication and Exploitation Procedures

Disinformation fabrication and exploitation procedures are key operational procedures in implementing the SCAC attack. In Section 3.2.2, four techniques are listed to generate disinformation used in smart grid security. Based on these general approaches, detailed procedures are needed to realize the attack objective.

Generally speaking, disinformation spreading can be developed as a service offered in underground and gray marketplaces. For example, some companies can abuse social media to damage another organization’s reputation, posting misleading messages by “influential” accounts with thousands of followers. The influence is great. One false tweet may reach more than 100 million people, and multiple coordinated accounts to spread disinformation could have more devastating consequences.

As discussed before, disinformation with different contents can result in different levels of impact. When an attacker generates a piece of disinformation, he/she should consider how to achieve a state transition that can lead to the physical system’s instability or disruption. In this paper, we propose two methods of disinformation generation for the Drastic Demand Change (DDC) manipulation threat. The DDC attack can be used in the smart grid, where the demand is the load requirement of power users; for other CPS infrastructures, the demand may be other user inputs that are fed into the system and need processing. We believe that generally, the larger the difference between the actual and the modified demand, the longer the time to regulate and restore the state of the physical system. The result of numeric simulation is shown in Section 5.2.3. The two kinds of methods are described as follows.

#### 3.3.1. Disinformation Based on the Fast Attack

Because attackers need to achieve a large disparity between the original demands and current demands, they should try to manipulate more users and abruptly change the demands of users. In this way, disinformation generation based on the fast attack is proposed.

Disinformation based on the fast attack refers to a piece of disinformation that can abruptly change the total demands when a large number of users have been controlled. This process can be described by (Equation 1).
(1)$$\begin{array}{cc} \; & N_{o}\left(t\right)=0\left(t<T\right)\\ \; & N_{o}\left(T\right)=N_{e}\left(T\right) \end{array}$$
where $N_{e}\left(T\right)$ represents the number of users who believe in the disinformation at time *T* and $N_{o}\left(t\right)$ denotes the number of users who execute operations based on the content of the disinformation at time *t*.

An example of a fast attack is shown in Figure 8. We describe the evolution of the number of users who believe in disinformation (i.e., controlled users) and the number of users who change their demands based on disinformation (i.e., users who launch attacks passively). Disinformation is propagated on social media at *t* = 0. The number of users who believe in the disinformation increases over time, and the number starts to increase drastically from t=5.

#### 3.3.2. Disinformation Based on Reverse Demands Attack

Sometimes, a piece of disinformation may not cause enough demand changes. The attackers can spread multiple pieces of disinformation to change demands continuously. In this way, the reverse demands attack based on multiple pieces of disinformation is proposed.

In this attack mode, some pieces of disinformation are generated first to gradually change the demands of users in one direction and then abruptly change the demands in the opposite direction, which can effectively utilize the demand-response idea to achieve the larger attack impact. We use an attack sequence Ts = $\left\{ts_{1},ts_{2},\cdots ,ts_{N}\right\}$ to describe the attacks from multiple pieces of disinformation. Attack action $ts_{i}$ means an attack of the *i*th piece of disinformation. Ts can be divided into two parts: gradually change demands; and abruptly change demands in the opposite direction.

Gradually change demands: From attack action $ts_{1}$ to $ts_{N-1}$, attackers gradually change the demands of users in one direction. During the process, the changing demands should not trigger any alert, and the system keeps stable under new demands.Abruptly change demands in reverse: When an attacker has controlled a large number of users, action $ts_{N}$ would drastically change the demands of users in the opposite direction from the impact of $ts_{N-1}$. For example, when attack action $ts_{N-1}$ decreases the demands of users, action $ts_{N}$ increases the demands of users.

Formula (Equation 2) is used to describe the process:(2)$$\begin{array}{cc} \; & Dem_{ts_{i}}\left(DN_{x}\left(t_{i}\right)\right)\times Dem_{ts_{j}}\left(DN_{x}\left(t_{j}\right)\right)>0\;i,j<N\\ \; & Dem_{ts_{i}}\left(DN_{x}\left(t_{i}\right)\right)\times Dem_{ts_{N}}\left(DN_{x}\left(t_{N}\right)\right)<0\\ \; & N_{o}\left(t_{N}\right)=N_{e}\left(t_{N}\right) \end{array}$$
where $t_{i}$ means the time that the *i*th attack action $ts_{i}$ occurs, $DN_{x}\left(t_{i}\right)$ means the demands of users who have been controlled by the attack action $ts_{i}$, and function $Dem_{ts_{i}}\left(\right)$ means the impact from attack action $ts_{i}$ for controlled demands. Figure 9 shows an example to illustrate the reverse demands attack, where the relationship between attack actions and demands is described. Demands are decreased gradually first, and the monitors of the CPS do not regard the changes in demands as exceptions. At the same time, the state of the physical system is regulated based on new demands. Until the 10th attack action, the attacker has controlled a large number of users and abruptly modifies the direction of change in demands. The total demand increases drastically such that the state of the physical system changes to the limit of violating the laws of physics.

## 4. Formal Description and Evaluation of the SCAC Model

In this section, we describe the SCAC model in a relatively formal way. The formal description helps the readers understand it more deeply and grasp the key points of the attack model. First, we use the industrial control system framework to describe and analyze the problem of physical system instability; then, we describe and evaluate how the disinformation infects users and finally disrupts the physical system.

### 4.1. Formal Description of the Physical System Instability Mechanism

There are many mechanisms and factors that cause system instability. In the case of industrial power systems, the most common disturbances that produce instability are: short circuits, switching operations, loss of a tie circuit to a public utility, loss of a portion of on-site generation, starting a motor that is large relative to the system’s generating capacity, impact loading on motors, and an abrupt decrease in electrical load on generators. From the above, we can see that load demand change is an important factor in disrupting the power grid. Generally speaking, a large-scale CPS usually provides service to numerous users. When user demands vary drastically, the system stability may be harmed.

We first introduce the physical system model that responds to user demand changes by using a seven-tuple and then describe the DDC threat proposed as an efficient method to realize SCAC in Section 3.3. Figure 10 shows the process of the stability control of the physical system. It works as a feedback process, during which the system responds to demand changes iteratively. In the case of a smart grid, when user demands have a change in voltage or electricity current, actuators can automatically respond to the change, and new load requests lead to the imbalance between the original supply and new demands. After that, the system state may have a transition. In the third step, sensory data that describe the system state are sent to the controller. The controller estimates the system state based on sensory data in the fourth step. In the fifth step, the controller issues commands to regulate the system state. Traditionally, the duration of the regulation from one state to another state has a threshold. The threshold is decided based on the laws of physics. When the duration is longer than the threshold, some components may be disconnected from the physical system or even disrupted.

The process can be modeled by the seven-tuple:(3)P(E,S,ΔD,C,W,T,R)
where:

$E=\{e_{1},e_{2},e_{3},\cdots ,e_{n}\}$ is a finite set of estimated states, where *n* describes the number of estimated states.

$S=\{s_{1},s_{2},s_{3},\cdots ,s_{m}\}$ is a finite set of real states of the physical system, where *m* describes the number of real states.

$\Delta D=\{\Delta d_{1},\Delta d_{2},\Delta d_{3},\cdots ,\Delta d_{x}\}$ is a set of the changed demands of users, where $\Delta d_{i}$ denotes the *i*th kind of changed demands of users.

$C=\{c_{1},c_{2},c_{3},\cdots ,c_{n}\}$ is a finite set of control commands, where $c_{i}=\{cmd_{1},cmd_{2},$$cmd_{3},\cdots ,cmd_{y}\}$ is a set of commands that respond to the estimated state $e_{i}$.

$W=\{\cdots ,w\left(\Delta d_{i},s_{j},s_{k}\right),\cdots \}$ is a set of state transitions, where $w(\Delta d_{i},s_{j},s_{k})$ means that when the current state of the physical system is $s_{j}$ and the changed demand is $\Delta d_{i}$, the state of the physical system would become $s_{k}$.

$T=\{\cdots ,t_{ij},\cdots \}$ is a set of the time cost, where $t_{ij}$ means the time cost that the state $s_{i}\left(e_{d}=s_{i}\right)$ is regulated to the state $s_{j}$ under the set of commands $c_{d}$.

$R=\{\cdots ,r\left(s_{i},s_{j},t_{limit}\left(ij\right)\right),\cdots \}$ is a finite set of relationships, where $t_{limit}\left(ij\right)$ means the limited time cost during which the state $s_{i}$ must be converted to the state $s_{j}$. If time cost $t_{ij}$ is longer than $t_{limit}\left(ij\right)$, it means that the laws of physic are violated and the physical system would be disrupted.

Based on the seven-tuple in (Equation 3), we describe the DDC threat. When demands have a great change, the physical system would try to provide services under demand-response services. The changed demand $\Delta d_{i}$ leads to the state transition of the physical system from $s_{j}$ to $s_{k}$, and then, $s_{k}$ needs be regulated to the state $s_{l}$; however, the time cost $t_{kl}$ may be longer than $t_{limit}\left(kl\right)$, which can be described by (Equation 4). Under this situation, the destruction of the physical system occurs.
(4)$$\begin{array}{cc} \; & w(\Delta d_{i},s_{j},s_{k})\in W\\ \; & w(0,s_{k},s_{l})\in W\\ \; & r(s_{k},s_{l},t_{limit}\left(kl\right))\in R\\ \; & t_{kl}>t_{limit}\left(kl\right) \end{array}$$

Although it is hard for users to have the opportunity to manipulate the controller, the DDC threat is not so difficult to implement from social domains. In Section 3.3, the fast attack and reverse demands attack are specific forms of such attacks. Malicious people attack users in an effort to drastically change their demands, satisfying (Equation 4). There mainly exist two tactics to achieve DDC: changing the demands of a large number of users; and greatly changing a small part of the users’ demands.

### 4.2. Formal Evaluation of the SCAC Attack Effect

In this subsection, we describe how to evaluate the impact of the SCAC attack in detail after attackers have generated the disinformation. We first search social relationships among users and then evaluate the number of manipulated users who execute operations according to the disinformation. Third, based on the result from the previous two steps, we evaluate demand changes and judge whether the disinformation can cause the disruption of the physical system. The process is described as follows.

#### 4.2.1. Search and Analyze the Social Relationships

Social relationships refer to correlations among users. A correlation between two users denotes that two users are familiar and a piece of disinformation can propagate between them. On different social media platforms, there may be different social relationships. Previous works [33,35] have demonstrated that different structures of the relationship among users have a different influence on information propagation on social media. Therefore, we first need to get the social relationships among users on the corresponding social media. Information on social media is often public, and attackers can analyze the relationships of users based on the collected information.

#### 4.2.2. Estimate the Number of Infected Users

To evaluate the number of infected users, we need to know how disinformation propagates on social media. We use the model from [33] to evaluate the number of manipulated users over time, which is described as follows: There exist three roles in the process of disinformation propagation: believer, fact-checker, and susceptible users. A believer believes in the disinformation and may tell friends that the information is true. A fact-checker does not believe in the malicious information and may tell friends that the information is false. A susceptible user is neutral.

The *i*th user has three states: $\left\{s_{i}^{B}\left(t\right),s_{i}^{F}\left(t\right),s_{i}^{N}\left(t\right)\right\}$. When the *i*th user is a believer at time *t*, $s_{i}^{B}\left(t\right)=1$, or otherwise, $s_{i}^{B}\left(t\right)=0$. When the *i*th user is a fact-checker at time *t*, $s_{i}^{F}\left(t\right)=1$, or otherwise, $s_{i}^{F}\left(t\right)=0$. When the *i*th user is a susceptible user at time *t*, $s_{i}^{N}\left(t\right)=1$, or otherwise, $s_{i}^{N}\left(t\right)=0$. The state transition is shown in Figure 11. Figure 11 quantitatively models the social user state transition when receiving the disinformation, while Figure 7 is a descriptive process model of user behavior in implementing attack procedures.

Next, the related parameters are computed as follows: Symbol $p_{i}\left(t\right)$ denotes the probability mass function of the *i*th user at time *t*:(5)$$p_{i}\left(t\right)=\left[p_{i}^{B}\left(t\right),p_{i}^{F}\left(t\right),p_{i}^{N}\left(t\right)\right]$$

States $s_{i}^{B}\left(t\right),s_{i}^{F}\left(t\right),s_{i}^{N}\left(t\right)$ can be computed in (Equation 6).
(6)$$\begin{array}{cc} \; & s_{i}^{B}\left(t\right)=\left\{\begin{array}{cc} 1\; & \mathrm{If}\;Random\left(0,1\right)\leq p_{i}^{B}\left(t\right)\\ 0\; & \mathrm{Else}\;\;\;\;\;\;\;\;\; \end{array}\right.\\ \; & s_{i}^{F}\left(t\right)=\left\{\begin{array}{cc} 1\; & \mathrm{If}\;p_{i}^{B}\left(t\right)<Random\left(0,1\right)\leq p_{i}^{B}\left(t\right)+p_{i}^{F}\left(t\right)\\ 0\; & \mathrm{Else}\;\;\;\;\;\;\;\;\;\;\;\;\;\;\;\;\; \end{array}\right.\\ \; & s_{i}^{N}\left(t\right)=\left\{\begin{array}{cc} 1\; & \mathrm{If}\;Random\left(0,1\right)>1-p_{i}^{N}\left(t\right)\\ 0\; & \mathrm{Else}\;\;\;\;\;\;\;\;\;\;\;\; \end{array}\right. \end{array}$$
where function Random(0,1) generates a random value in (0, 1) and functions $p_{i}^{B}\left(t\right)$, $p_{i}^{F}\left(t\right)$, and $p_{i}^{N}\left(t\right)$ are computed based on the spreading rate, the credibility of the disinformation, and the state transition possibilities in Figure 11.

At time *T*, the evaluated number of users who are controlled by disinformation, ${\hat{N}}_{e}$, can be computed:(7)$${\hat{N}}_{e}=\sum_{i=1,2,3\cdots ,n}s_{i}^{B}\left(T\right)$$
where *n* denotes the number of users.

#### 4.2.3. Estimate Demand Change When Manipulated

We first describe the real demands at time *T* after a piece of disinformation is disseminated at Time 0 and some users execute operations based on the disinformation. Then, we show how to evaluate demands when manipulated.

The demands of the *i*th user at time *T*, $d_{i}\left(T\right)$, can be computed:(8)$$d_{i}\left(T\right)=M\left(o_{i}\left(T\right)\right)$$
where function M() means the modified demand after the user believes in the disinformation and $o_{i}\left(t\right)$ means the demand when the user is not infected.

The total demands $D_{dem}\left(T\right)$ at time *T*:(9)$$D_{dem}\left(T\right)=\sum_{i\in Believers}M(o_{i}\left(T\right))+\sum_{i\notin Believers}o_{i}\left(T\right)$$

Although attackers can evaluate the number of infected users at time *T*, they cannot know which specific users are controlled. Moreover, different users have different demands. Therefore, Equation (Equation 9) cannot be obtained by the attackers. We use Equation (Equation 10) to evaluate total demands $\hat{D_{dem}}\left(T\right)$ at time *T*:(10)$$\hat{D_{dem}}\left(T\right)=\left(M\left(\frac{\sum o_{i}\left(T\right)}{n}\right)-\frac{\sum o_{i}\left(T\right)}{n}\right)\times {\hat{N}}_{e}+\sum o_{i}\left(T\right)$$
where *n* denotes the total number of users.

Once the demands at time *T* have been estimated, the attackers can evaluate the current state of the physical system based on the characteristics of the CPS, which needs related domain knowledge. Because the adversaries who attack critical infrastructures often have high intelligence and ability, it is possible that they can know the values of the parameters and evaluate state transitions based on new demands. After that, attackers can also evaluate whether this disinformation attack can disrupt the physical system represented by (Equation 3).

## 5. Simulation Analysis of SCAC in a Smart Grid

In this section, the case of a power grid is presented to illustrate the feasibility and effectiveness of SCAC. We first introduce the system model and power price model, and then, we use simulation experiments to demonstrate the attack’s effects.

### 5.1. Power System Model and Power Price Model

In the power grid, the frequency should remain 50 Hz (or 60 Hz in some areas). Only a 1% deviation from a normal frequency may damage equipment and infrastructure and will result in automatic load shedding or other control actions to restore system frequency. Over a period of time $T_{interval}$, when the deviation between the current frequency and normal frequency is larger than the threshold $\Delta f_{max}$ for some time $T_{thr}$, generators can be disconnected from the power grid, and a disruption may occur [11,36,37].

Figure 12 depicts a simplified system model of the power grid, which is comprised of the central controller, direct load controller, generators, and demands of users. The central controller regulates the generators and the direct load controller based on the frequency. When demands are not equal to generation, the frequency fluctuates continuously. A serious fluctuation can cause the switch connected to the generators to turn off. When the frequency is higher than 50 Hz, the central controller first instructs the direct load controller to try to increase the use of power. If the frequency is still higher than the normal frequency, the output from the generators is decreased. Conversely, when the frequency is lower than the normal frequency, the controller first regulates the direct load controller to try to decrease the use of power.

As described in [36], Formula (Equation 11) shows the relationship among frequency, demands, and generation.
(11)$$\left\{\begin{array}{cc} \; & P_{G}\left(t+d_{T}\right)=P_{G}\left(t\right)+\left(P_{TAR}\left(t\right)-P_{G}\left(t\right)\right)\times M\times d_{T}\\ \; & P_{TAR}\left(t\right)=\frac{f_{sp}-f\left(t\right)}{0.04\times f_{nor}}\times P_{GMAX}\\ \; & I=\frac{2\times P_{GMAX}\times H}{\omega_{NOM}^{2}}\\ \; & \frac{1}{2}\times \;I\times \omega {\left(t\right)}^{2}=\frac{1}{2}\times I\times \omega {\left(t-d_{T}\right)}^{2}+P_{S}\left(t\right)\times d_{T}\\ \; & \omega (t)=2\pi \times f(t)\\ \; & P_{S}\left(t\right)=P_{G}\left(t\right)-P_{L}\left(t\right) \end{array}\right.$$
where $P_{G}\left(t\right)$ means the generation of the generators at time *t*, $P_{L}\left(t\right)$ denotes the load demands of the users at time *t*, $d_{T}$ means a unit of time, and n$P_{GMAX}$ and $\omega_{NOM}$ denote the largest power from the generators and the rotating frequency at the normal frequency $f_{nor}$, respectively. *H* is a constant, which is often equal to 4 s. *M* is equal to 0.3. $f_{sp}$ means the generator’s set point in Hz.

Power price model: Previous works [38,39] demonstrated that user demands can be changed by modifying the power price. The relationship between changed power price ΔPrice and changed demands Δdemands can be described by a linear function in Equation (Equation 12), which was used in [38]. *e* and *f* are constant parameters.
(12)$$\begin{array}{c} e\times \Delta Price+f=\Delta demands \end{array}$$

### 5.2. Simulation Evaluation

The evaluation has three objectives: (1) illustrate the effectiveness of the social collective behavior attack based on disinformation propagation on social media; (2) show the importance of generating proper disinformation and evaluating the impact of the attack; (3) validate the effectiveness of disinformation generation based on the fast attack and gradually the reverse demands attack.

We used Java to simulate the power grid system and began by setting the basic parameters and introducing two kinds of attacks. Traditionally, total demands for electric power in the next two days are similar [39]. Attackers can get total demands under the normal situation from public data. We used the ERnetwork structure [40] to describe the social relationships of users on social media. There were 10,000 users with different demands. The demands of users were randomly allocated. We used the state transition model in Figure 11 to simulate disinformation propagation, where $p_{forget}$ = 0.3, $p_{verify}$ = 0.1, and β = 0.1 [35]. Credibility α can be changed based on the contents of the disinformation. In power grid model, as described in [36], $P_{GMAX}$ = 4800 MW, $f_{SP}$ = 51 Hz, $d_{T}$ = 1 s, $\Delta f_{max}$ = 0.2 Hz, $T_{interval}$ = 60 s, and $T_{thr}$= 20 s. At the same time, the direct load controller can increase or decrease the 200 MW load. For the power price model, we set *e* = −1000 and *f* = 0. The real power price is shown in Figure 16a.

One piece of disinformation is generated as a notification message, which tells users to stop using power at 8:00 p.m. When a user believes in this piece of disinformation, the demand *D* changes as *D* × 0.2. Next, we introduce two kinds of attacks: Attack Type I and Attack Type II, which are combinations of the disinformation fabrication approaches in Section 3.2.

Attack Type I: attackers only propagate the false notification on social media.Attack Type II: besides the false notification, attackers propagate false price messages, which can gradually modify the power price such that everyone eventually believes in the false prices.

#### 5.2.1. The Influence of Disinformation Contents on the Attack Effect

A larger value of α means that attackers generate the disinformation with higher credibility. In Figure 13, we show the impact of the attack when α is equal to 0.2 and 0.8, respectively.

In Figure 13a,b, the process of disinformation propagation is shown. We can see that the number of believers converges. A larger α can obtain a larger number of believers. We show the demands of users under two kinds of situations in Figure 13c. A larger α can achieve greater fluctuations in total demand. When the propagation reaches the stable state and α is equal to 0.2, we can see that the demands have a very small change. When α is equal to 0.8, the total demands have a bigger change. In Figure 13d, we show the changes of frequency under two kinds of demand changes. From this result, we can clearly see that when the piece of disinformation with α = 0.8 is propagated among users, total demands change from 2400 MW to 1445 MW when users have executed operations based on the disinformation at 8 p.m. After that, the frequency starts deviating, and the duration for which the deviation of frequency exceeds the threshold 0.2 Hz is longer than 20 s. The result leads to the disconnection of generators. When α is equal to 0.2, the generators are not disconnected. The situation of α = 0.8 illustrates that the social collective attack by propagating disinformation on social media is effective when an attacker can generate the proper disinformation.

Comparing the attack effect of two different values of α, we can know that the contents of disinformation can greatly influence the impact of the attack.

#### 5.2.2. The Accuracy of Impact Evaluation

In this experiment, we validate the accuracy of the impact evaluation in the disinformation attack. Considering that every time the simulation experiment under the same attack may achieve different levels of impact because of the randomness of disinformation propagation, we ran every experiment 1000 times under the same parameter α and got the average demands that were seen as real demands under the disinformation attack.

In Figure 14, we show the duration for which the deviation of the frequency exceeds the threshold in 60 s under real demand changes and estimated demand changes with respect to α. Comparing the attack effects of real demands and evaluated demands under the fixed α, we can see that except for α = 0.5, the evaluation of the disinformation attack can relatively accurately predict whether a piece of disinformation with a fixed parameter α can cause instability. When α is equal to 0.5, the disinformation attack can cause the new demands to be in the range [1600 MW, 1700 MW]. In Section 5.2.3, we show that when demands are changed from 2400 MW to *d*(*d* ∈ [1600 MW, 1700 MW]), the duration for which the deviation of the frequency exceeds 0.2 Hz has a drastic change. Therefore, a small evaluated error between real demands and evaluated demands would lead to a significant difference. In summary, for most of the situations, our evaluation method is accurate to reveal whether the instability of the power grid occurs due to the corresponding disinformation attack.

#### 5.2.3. The Effectiveness of the Fast Attack and the Reverse Demands Attack

In this experiment, we illustrate the effectiveness of the two methods of the DDC threat proposed in Section 3.3 by showing the relationship between demand changes and the impact of the attack.

In Figure 15, we show the relationship among the initial demands, decreased demands, and the duration for which the deviation of the frequency exceeds the threshold. The initial demands refer to the demands before the demands of users are manipulated. Decreased demands mean the demands that are decreased by users because of social collective attacks. For the fixed initial demands, with the increase in the number of decreased demands, the color tends to be red, which means that a larger change in demands can have a greater impact. For the fixed number of demand changes, we can see that under different initial demands, the color is the same, which means that the initial demands are not directly related to the impact.

Next, we illustrate that reverse demand attacks mainly based on Attack Type II can achieve a better impact than fast attacks mainly based on Attack Type I. The parameter α of this piece of disinformation is 0.5. In Figure 14, we see that sometimes, Attack Type I with α = 0.5 cannot cause the disconnection of generators. In Figure 16, we show the impact of Attack Type II with α = 0.5. Attackers modify the power price from 18:50 to 20:00. Figure 16a describes the real prices and modified prices over time. Figure 16b describes the demand changes for real prices and the modified price, respectively. Compared with the demands under the normal price, the demands of users under power price modification attacks gradually increase. During the process, the frequency of the power grid keeps invariant.

Before 8:00 p.m., the modified power price can cause the changes of total demands from 2400 MW to 3000 MW. When infected users execute the operation based on the disinformation with α = 0.5 at 8 p.m., the believers completely turn off the power under the influence of the changed power price. The attack effects of the two kinds of attacks are shown in Figure 15c. Compared with Attack Type I, Attack Type II can achieve a larger deviation of the frequency. The impact of Attack Type II shows that generators would be disconnected from the power grid, which illustrates that reverse demand attacks can increase the demand change to have a better impact.

## 6. Conclusions

In this paper, we focused on one new kind of attack for large-scale CPSs called Social Collective Attack on CPS (SCAC). The biggest difference between the social collective attack and previous attack methods is that the former manipulates collective behavior to disrupt the CPS by directly intruding into the social domain. We describe an implementation of the social collective attack based on disinformation propagation on social media. We analyzed whether the changes of collective behaviors can disrupt the CPS and how the attackers launch disinformation attacks to better manipulate collective behavior. Our work demonstrates that the social collective behavior attack is possible and harmful.

These attacks may undermine the system’s ability to provide mission-critical services. Defenders of large-scale CPSs should consider this kind of attack. Since these attacks may not leave signs in the information system and the physical system before the CPS is disrupted, we will combine the behavior of users with information on social media to propose an effective defensive method, which will be our next concern.

## Figures and Tables

**Figure 1 sensors-21-00991-f001:**
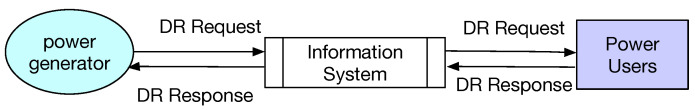
Demand response model.

**Figure 2 sensors-21-00991-f002:**
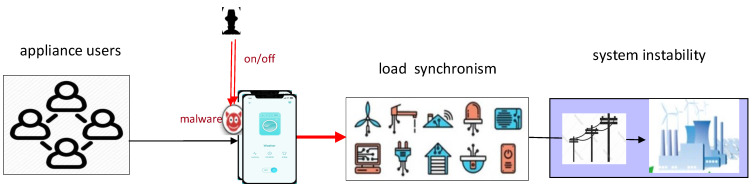
Botnet and system instability: attack from the cyber domain.

**Figure 3 sensors-21-00991-f003:**
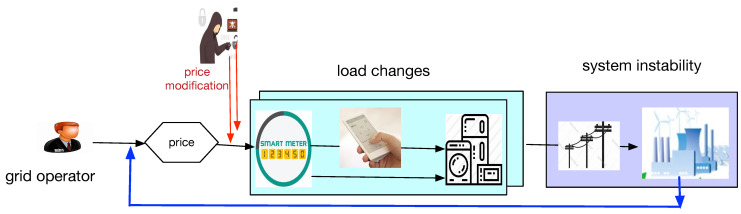
Price modification and system instability.

**Figure 4 sensors-21-00991-f004:**
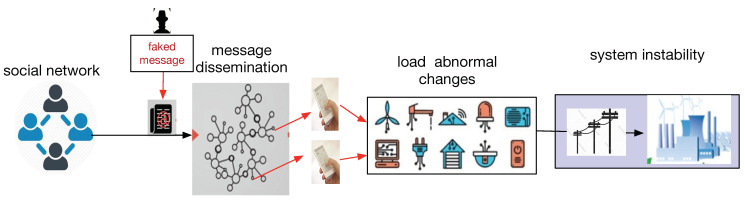
Social-cyber-physical: attack initiated from the social domain.

**Figure 5 sensors-21-00991-f005:**
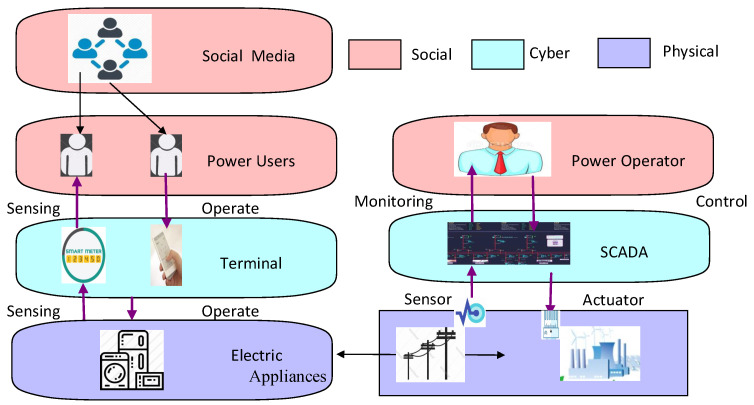
Cyber-Physical System (CPS) model with social domains.

**Figure 6 sensors-21-00991-f006:**
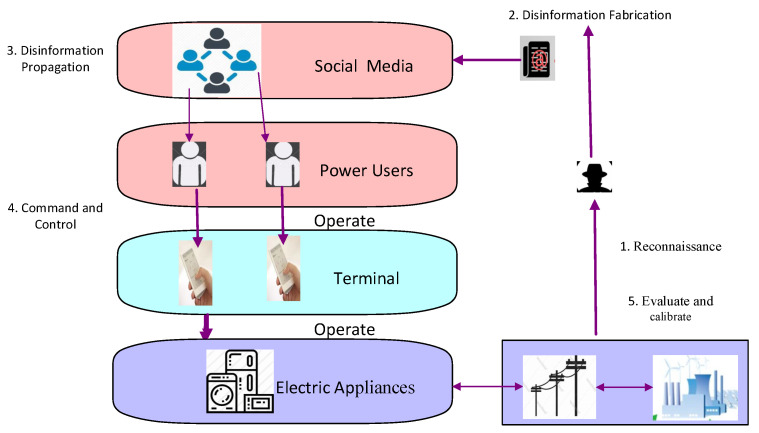
Steps of Social Collective Attack on CPS (SCAC).

**Figure 7 sensors-21-00991-f007:**
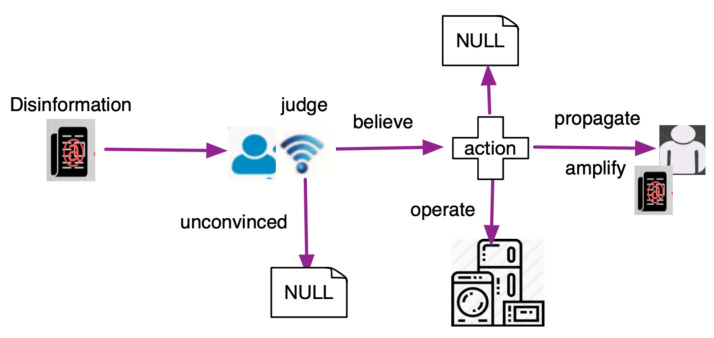
User behavior model in SCAC.

**Figure 8 sensors-21-00991-f008:**
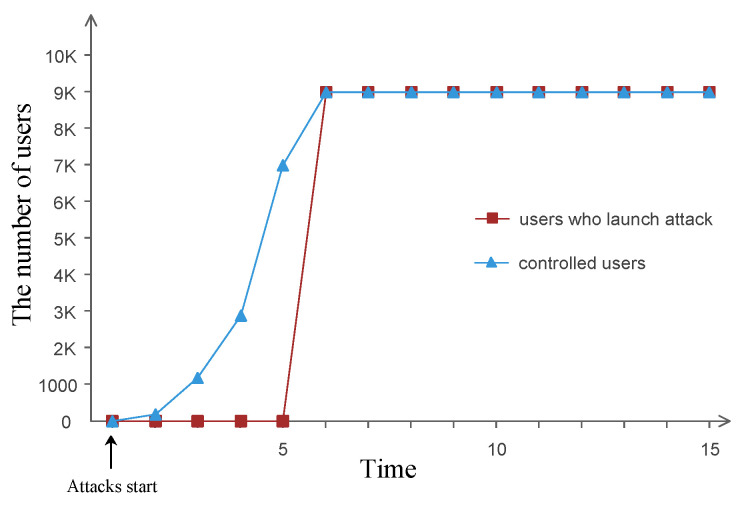
An example of a fast attack.

**Figure 9 sensors-21-00991-f009:**
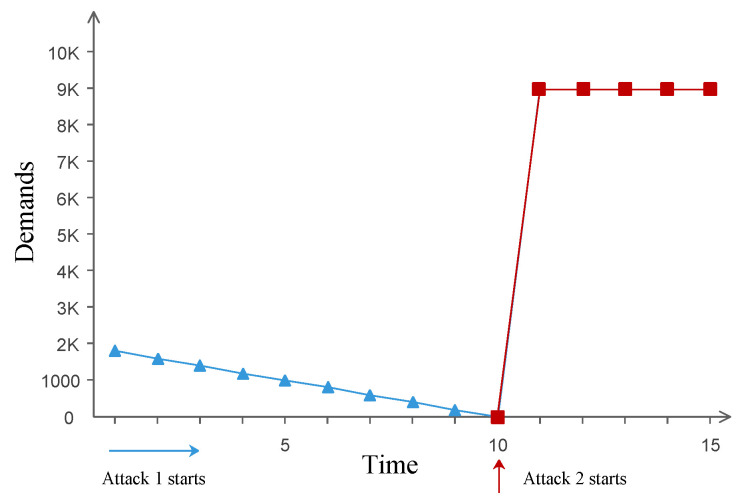
An example of the gradual reverse demands attack.

**Figure 10 sensors-21-00991-f010:**
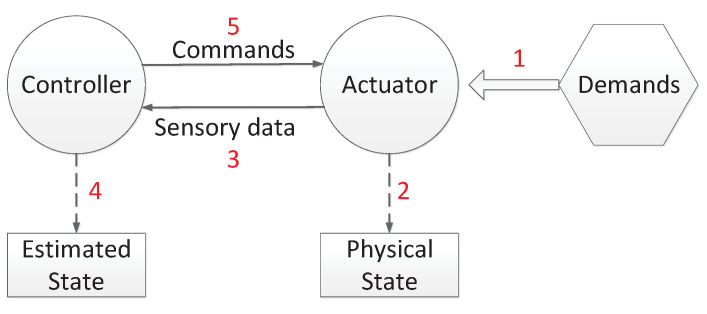
Process of the stability control of the physical system.

**Figure 11 sensors-21-00991-f011:**
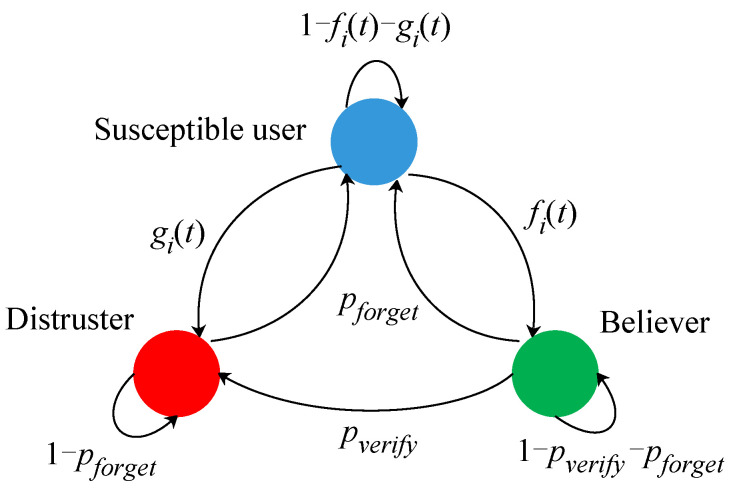
State transition model [33].

**Figure 12 sensors-21-00991-f012:**
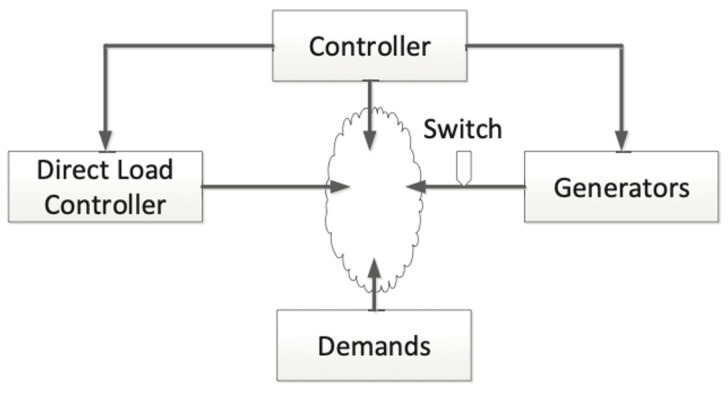
A simplified model of the power grid system.

**Figure 13 sensors-21-00991-f013:**
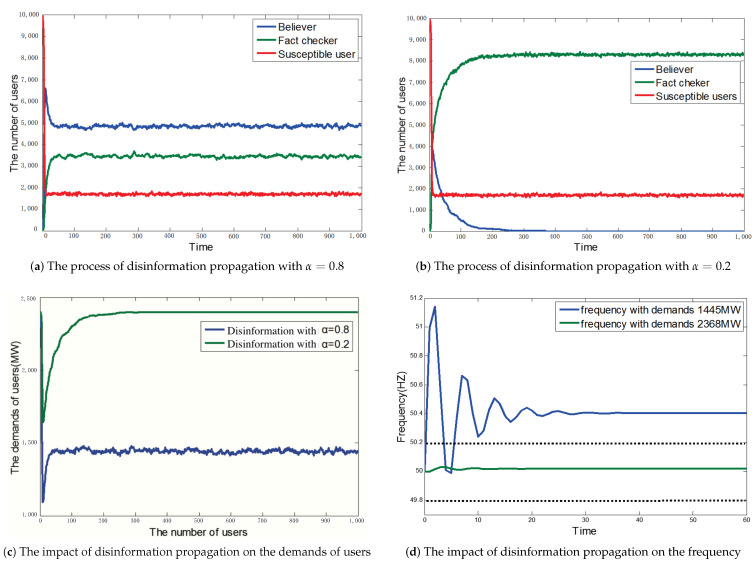
Comparing the effects of the disinformation attack with α=0.8 and with α=0.2, where β = 0.5 and initial demand is 2400 MW.

**Figure 14 sensors-21-00991-f014:**
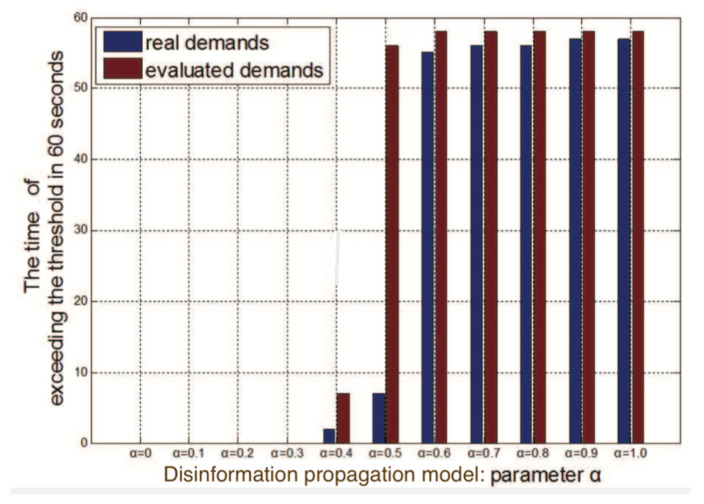
Attack effects of real changing demands and evaluated changing demands with the change of parameter α.

**Figure 15 sensors-21-00991-f015:**
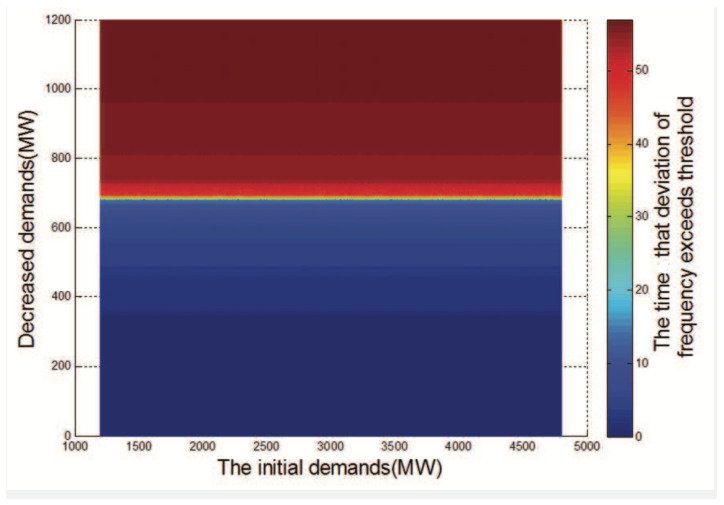
The relationship among initial demands, changed demands, and the attack effect.

**Figure 16 sensors-21-00991-f016:**
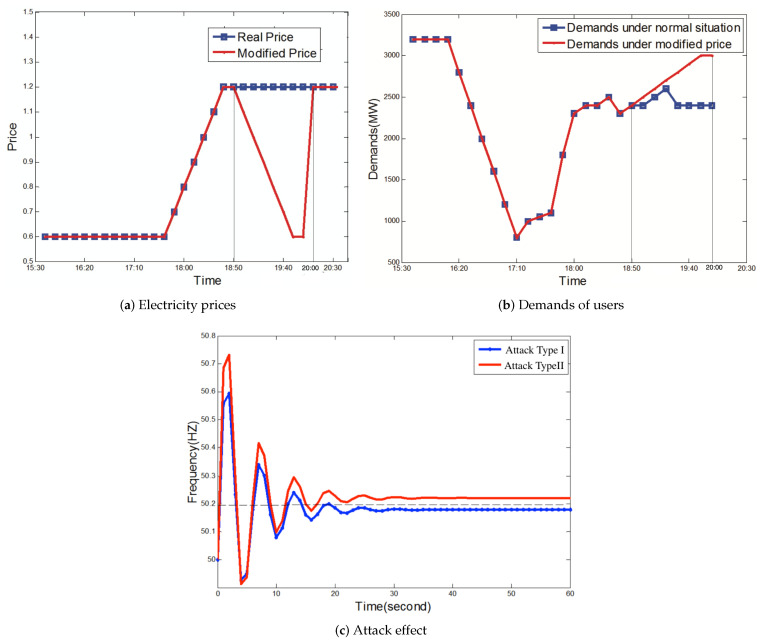
Comparing the attack effects between Attack Type I and Attack Type II, where α = 0.5 and initial demands = 2400 MW.

## Data Availability

Data sharing not applicable.

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
