# Peer review of "Social Collective Attack Model and Procedures for Large-Scale Cyber-Physical Systems"

_sensors, 2021, doi:10.3390/s21030991_

Round 1

Reviewer 1 Report

The contribution is not significant. There are serious flaws in the manuscript such as linkage among sections. The paper is not focused on the main contribution. The presentation os some figures is poor. For example, Figure 8 and Figure 9.

Reviewer 2 Report

This paper is to study Social collective attack models and procedures for large-scale cyber-physical systems. It looks exciting, however, the authors have to update following next;

  1. what is large-scale? Authors have to define 'large-scale in body of the paper.
  2. New attack model that the authors said this paper describe, however, not clear what new attack model is.
  3. In section 3.1, there are physical domain, cyber domain, social domain. But for mission or purpose and so on, why they were defined are not clear.
  4. What difference between 'attack' and 'approach'? The title of 'section 3.2.1. price-based attack,' '3.2.2 incentive-based attack', on the other hand, '3.2.3 Lose-avoidance approach'. I think 'attack' and 'approach' are differences. If each has difference meaning, I suggest that two parts should be separated.
  5. in figure 11, background or this figure should be white color like any other figures.
  6. descriptions and analysis for the attacks are not enough.

Round 2

Reviewer 1 Report

The authors have addressed my comments.

Author Response

Thank you for your time and effort in reviewing our revised manuscript.

Reviewer 2 Report

This presented a social collective attack model and procedures for large-scale cyber-physical systems. It looks interested area, but because of next reasons, I decided to major revision.

  1. No impact in the abstract. The abstract contains the existing problem, new idea and effect such as strong points, however these descriptions are so general.
  2. In Related work section, remove general sentences. Fill direct or close works to this paper’s direction, and too much than we expect.
  3. Hard to read and to understand, not clear in ‘the section 3. Attack model and procedures’
  4. There are some figures (fig. 10. 11) and equations, but not clear how to process interactively.
  5. Figure 12 is the simplified system model of the power grid (in page 15 / 512nd line), but the graphs looks the results. (And it mentioned the fig 12(a)…(d) in page 17). Which is correct?
